# The microbiome compositional and functional differences between rectal mucosa and feces

Xiao-Fei Yin,[1] Taoyu Ye,[2] Han-Lin Chen,[1] Junyan Liu,[2] Xue-Feng Mu,[1] Hao Li,[2] Jun Wang,[2,3,4] Yuan-Jia Hu,[1] Hongzhi Cao,[2,3,5] Wen-Quan Kang[1]

**ABSTRACT**    In recent years, most studies on the gut microbiome have primarily focused on feces samples, leaving the microbial communities in the intestinal mucosa relatively unexplored. To address this gap, our study employed shotgun metagenomics to analyze the microbial compositions in normal rectal mucosa and matched feces from 20 patients with colonic polyps. Our findings revealed a pronounced distinction of the microbial communities between these two sample sets. Compared with feces, the mucosal microbiome contains fewer genera, with Burkholderia being the most discriminating genus between feces and mucosa, highlighting its significant influence on the mucosa. Furthermore, based on the microbial classification and KEGG Orthology (KO) annotation results, we explored the association between rectal mucosal microbiota and factors such as age, gender, BMI, and polyp risk level. Notably, we identified novel biomarkers for these phenotypes, such as *Clostridium ramosum* and *Enterobacter cloacae* in age. The mucosal microbiota showed an enrichment of KO pathways related to sugar transport and short chain fatty acid metabolism. Our comprehensive approach not only bridges the knowledge gap regarding the microbial community in the rectal mucosa but also underscores the complexity and specificity of microbial interactions within the human gut, particularly in the Chinese population.

**IMPORTANCE**  This study presents a system-level map of the differences between feces and rectal mucosal microbial communities in samples with colorectal cancer risk. It reveals the unique microecological characteristics of rectal mucosa and its potential influence on health. Additionally, it provides novel insights into the role of the gut microbiome in the pathogenesis of colorectal cancer and paves the way for the development of new prevention and treatment strategies.

**KEYWORDS**  gut microbiome, metagenomic sequencing, rectal mucosa, KEGG Orthology, differential markers

The gut microbiome is closely associated with human health (1, 2), influencing various physiological processes, disease risks, and therapeutic effects (3, 4). Its complex interplay with the host has been the subject of intense research, especially in the era of precision medicine, where understanding those microbial communities has become a focal point of scientific inquiry (5–7). Recent studies have highlighted the potential role of the gut microbiota in the onset and progression of various diseases, particularly inflammatory and malignant gastrointestinal conditions. For instance, the interaction between the gut microbiome and the host's immune system, through mechanisms like Toll-like receptor (TLR) signaling and inflammasome sensing, has been identified as critical pathways influencing intestinal carcinogenesis (8, 9).

This intricate relationship between the gut microbiota and human health extends to colorectal conditions. The gut microbiota has been found to influence the development

Address correspondence to Hongzhi Cao, caohongzhi@icarbonx.com, or Wen-Quan Kang, kangkwq@163.com.

Xiao-Fei Yin and Taoyu Ye contributed equally to this article. The Author order was determined based on their contribution to the article.

The authors declare no conflict of interest.

See the funding table on p. 13.

of colorectal polyps, which are precursors to colorectal cancer (CRC). Modern lifestyle changes and environmental factors can alter the gut microbiota's composition and function, potentially impacting the development and progression of these polyps. Understanding these changes is vital, especially considering that most CRC cases are sporadic but often preceded by dysplastic benign polyps, known as adenomas. These adenomas can evolve into malignant forms, a progression termed the adenoma–carcinoma sequence (10).

At present, the global scientific community has yet to reach a consensus on the microbial signatures associated with CRC risk, which may be due to the differences in the technical methods employed across different studies and the diverse locations where samples are taken. Techniques have evolved from initial microbial isolation and cultivation to subsequent 16S rRNA amplicon sequencing and the latest metagenomic studies. 16S rRNA amplicon sequencing focuses on the composition and diversity of microbial communities, identifying microorganisms at the genus level but lacks functional information. On the other hand, metagenomic sequencing, with its high throughput and subjecting to less bias than 16S rRNA amplicon sequencing, provides both classification information of bacteria and functional gene data. With its higher taxonomic resolution, metagenomics can identify species and furnish comprehensive information, including the type, quantity, and function of microorganisms (11). However, due to its relatively high cost and complexity of analysis, most studies still prefer to use the more cost-effective 16S sequencing technology (12–14).

Since feces can be collected non-invasively and repeatedly, most previous studies about gut microbiome had been based on such specimen (15, 16). However, there is increasing evidence to suggest that there may be significant variations in microbial composition between the gut mucosa and feces (17). While feces are often deemed to be a substitute for gastrointestinal lumen contents, they may not accurately reflect the complex interactions that occur directly at the gut mucosal surface. In fact, it has been shown that fecal and mucosal-associated microbiota are two distinct populations (18). Furthermore, the fecal microbiota is not distributed equally within feces and has its own unique biostructure (19). A recent study found significant differences between bacteria, phages, host proteins, and metabolites in the intestines versus feces (20), meaning that the microbial communities in the gut have specific compositions and functions at different locations. The latest study suggests that since the microenvironment of different locations is different, different tissue sites, due to their environmental characteristics and physiological functions, should also have distinct microecological characteristics.

The intestinal mucosa is the primary site for nutrient absorption and the immune barrier in the body. It also serves as the main medium for direct interaction between the gut microbiota and the host. Therefore, the microbes residing here are closely related to intestinal nutrition and immune mechanisms. Its microbial characteristics are more closely related to body functions and disease risks compared with the gut microbiota in feces. Although there have been a few studies that have compared the fecal microbiota with the intestinal mucosal microbiota, they have limitations, such as a smaller sample size, the use of 16S sequencing technology (21–24). Therefore, considering the importance of the intestinal mucosal microbiota and the limitations of previous research, we conducted this study.

Here, in order to further elucidate and quantify the similarities and differences in composition and phenotype between the intestinal mucosal microbiome and fecal microbiome, we collected normal rectal mucosal tissues and fecal samples from 20 patients with colorectal polyps at risk of colorectal cancer. Subsequently, metagenomic sequencing techniques were employed to examine the 20 paired samples, followed by an in-depth data analysis and comparison.

## RESULTS AND DISCUSSION

### The microbiome composition of rectal mucosa and feces

To investigate the potential microbial community variability of the large intestine, we collected feces and normal mucosal biopsies from 20 adults with colorectal polyps: comprising 12 males and eight females with an average age of 55 years (Fig. 1A; Table S1). In order to exclude the influence of different parts, all biopsies were taken from the rectum. All samples were prepared for shotgun metagenomic sequencing, resulting in 5.76G (SD = 0.29G) and 5.93G (SD = 0.83G) raw data for mucosa tissue and feces samples, respectively. In following sections, we detailed how this data were utilized to explore the microbial compositions of the mucosal biopsy and feces, looking at both their differences and underlying connections.

To assess the microbial diversity of our samples, we first computed the gene richness of fecal and mucosal samples, as shown in Fig. S1. On average, the mucosa exhibited a 6,661 gene-richness which is about two orders of magnitude lower than the 539,475 genes found in feces. Next, we examined how sample type affected microbiome diversity. The Principal Coordinate Analysis (PCoA) plot of the fecal-mucosal data sets demonstrated that feces cluster separately from mucosal samples (Fig. 1B). This indicates a significant difference between fecal and mucosal microbiomes.

To further interpret these findings, we calculated the number of shared genera between the two sets of samples (Fig. 1C). Almost half of the genera are shared between the mucosal and fecal microbiomes, while the other half are exclusive to feces. Only a select few genera, namely *Cyanothece, Oenococcus*, and *Xanthobacter*, are exclusive to the mucosa. *Cyanothece* is a unicellular cyanobacterium known for its diazotrophic nature and oxygenic photosynthesis (25). *Oenococcus* is a Gram-positive member of the Lactobacillaceae family (26), while *Xanthobacter* is a Gram-negative bacterium from the Xanthobacteraceae family (27, 28). Neither of these genera is a common commensal of the human digestive tract, so their occurrence may be transient (29, 30). Despite the overlap of about 50% of genera, the fecal and mucosal microbiomes are significantly different as the fecal microbiome has a richer variety (Fig. 1B). This difference in richness might have implications for the biological properties and functional roles that these microbiomes play in the host's health.

Next, we conducted a taxonomic analysis of mucosal and fecal microbiomes to find out which species might be used as a biomarker to distinguish the two groups (Fig. 2). The top ten most abundant detected phyla, genera, and species are displayed in

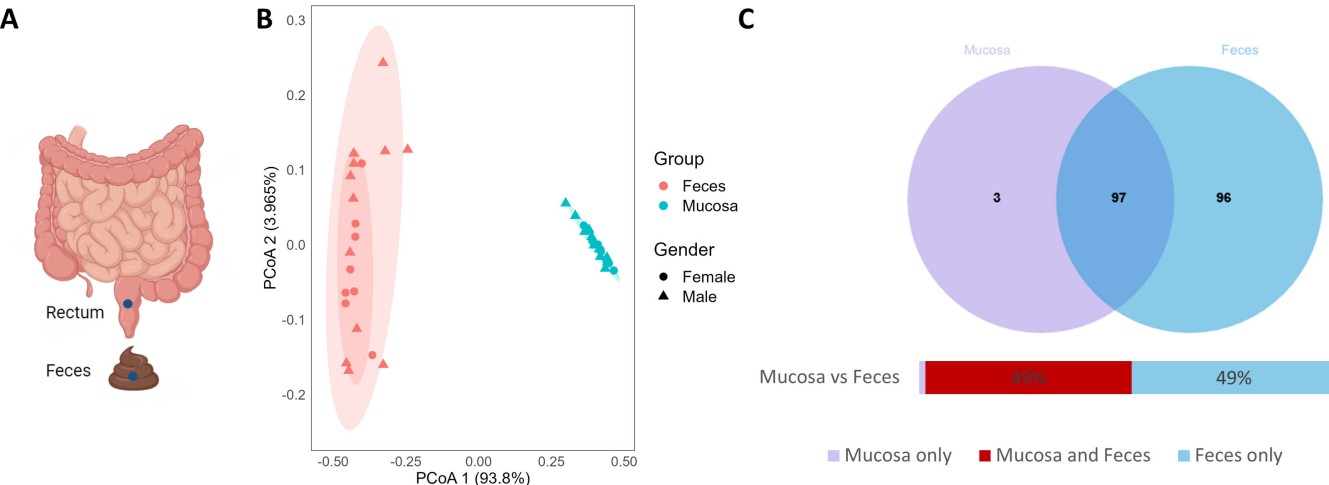

**FIG 1** Sampling locations and general overview of the fecal and mucosa-derived metagenomic data sets. (A) For each of the 20 enrolled subjects, biopsies of normal gut mucosa were collected from the rectum, and feces were also sampled. (B) PCoA plots of the downsized mucosa-feces data set at the phylum level, color-coded by sampling-location and shape-coded by gender. (C) Information on the genera of feces and mucosa. PCoA: Principal Coordinate Analysis.

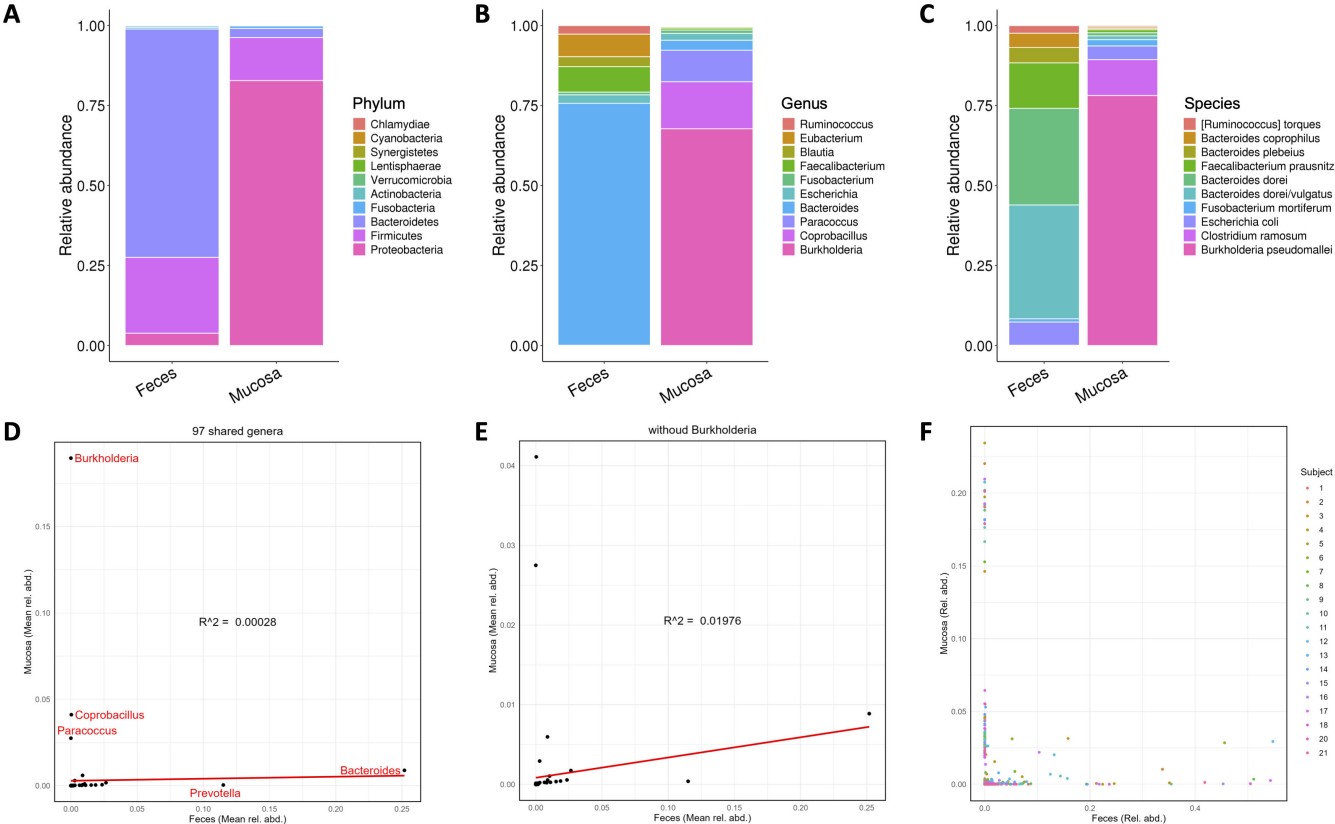

**FIG 2** Taxonomy of microbial communities from rectal mucosa and feces; *n* = 20 for each group. Top 10 most abundant phyla (A), genera (B), and species (C) among all samples. Phyla, genera, and species are sorted in the legends from the most to the least abundant in rectal mucosa. The corresponding feces values are also plotted for comparison in the same order. (D, E) Scatter plots of mean relative abundances of all 97 shared genera between mucosa and feces (D) and of the same genera except *Burkholderia* (E). (F) Scatter plots of the genera relative abundances of paired samples: Feces versus mucosa; data points are color-coded by subject.

(Fig. 2A through C), arranged in order from most to least abundant in the mucosa. Proteobacteria is the most dominant phylum in the mucosa and ranks third in feces. This dominance of Proteobacteria in the mucosa may be attributed to their adaptability in the mucosal environment. The mucosal surface, being in close proximity to the gut epithelial cells, often presents a unique set of challenges including immune responses and fluctuating oxygen levels (31). Furthermore, it is worth noting that in the context of intestinal diseases, Proteobacteria emerges as one of the most abundant phyla, containing many known human pathogens. Some studies have indicated that Proteobacteria is associated with various diseases, all of which are characterized by varying degrees of inflammation (32). Firmicutes was the second most represented phylum in both mucosa and feces, predominantly consisting of the genera *Coprobacillus* and *Faecalibacterium*, respectively. Notably, the percentage of Firmicutes was significantly more abundant in feces compared with mucosa. However, Bacteroidetes ranked third in mucosal tissue but first in feces, being the dominant phylum of feces (Fig. 2A). The higher abundance of Bacteroidetes in feces is likely due to their capability to degrade complex polysaccharides, which are abundant in the diet and reach the colon undigested (33, 34). Bacteroidetes possess a wide array of enzymes that allow them to break down these complex carbohydrates (35), making them particularly suited for the fecal environment. This aligns with previous research utilizing microdissection and sequencing techniques, which observed a predominance of Firmicutes and Proteobacteria in luminal crypts, with Bacteroidetes being poorly represented (36, 37). Compared with previous studies (20, 23), the novelty of our research lies in conducting metagenomic sequencing on both

the normal rectal mucosal tissue and feces of patients at risk for colorectal disease. This comprehensive approach allows us to gain a deeper understanding of the microbial composition and its potential role in colorectal health. By analyzing both mucosal and fecal microbiomes, we can identify specific microbial signatures that may serve as biomarkers for early detection of colorectal diseases.

To explore the co-variations of fecal and mucosal microbiome, we plotted the 97 genera that were shared by these two groups (Fig. 2D). The correlation between feces and mucosa was particularly low (R2 = 0.00028), which is suspected to be due to the enrichment of the *Burkholderia* genus in the mucosa (Fig. 2D). After eliminating this genus from the shared genera, the correlation between mucosa and feces became significantly stronger (R2 = 0.01976; Fig. 2E). Therefore, it can be concluded that the *Burkholderia* genus is the most relevant discriminating factor between feces and mucosa. These results are consistent with previous findings from mouse colon studies, which found that aerobic bacteria, particularly those belonging to the Burkholderiales, tend to dominate the crypt-specific core microbiota (37, 38).

## The significant microbiome difference between rectal mucosa and feces

In the comparison between rectal mucosa and feces, we identified a substantial number of differential features (*P* < 0.05), totaling 248. Specifically, six significant features were found at the phylum level, 70 at the genus level, and as many as 172 at the species level (see Tables S2 to S4 for details). From a biological perspective, the observed differences between the fecal and mucosal microbiomes can be attributed to the distinct environments and functions (39). The mucosal environment is exposed to different oxygen levels, immune responses, and nutrient availabilites compared with the fecal environment (31, 40). These factors can influence the colonization and abundance of specific microbial species (41).

Notably, the majority of these significant features showed higher relative abundance in feces. However, our primary interest lies in understanding the unique microbial landscape of mucosal tissues. According to Fig. 3, we identified 11 features with a higher relative abundance in the rectal mucosa. These encompassed one phylum (Proteobacteria), six genera (*Burkholderia*, *Coprobacillus*, *Paracoccus*, *Thauera*, *Vibrio*, *Lactobacillus*), and four species (*Burkholderia pseudomallei*, *Clostridium ramosum*, *Vibrio cholerae*, *unclassified Thauera sp.MZ1T*).

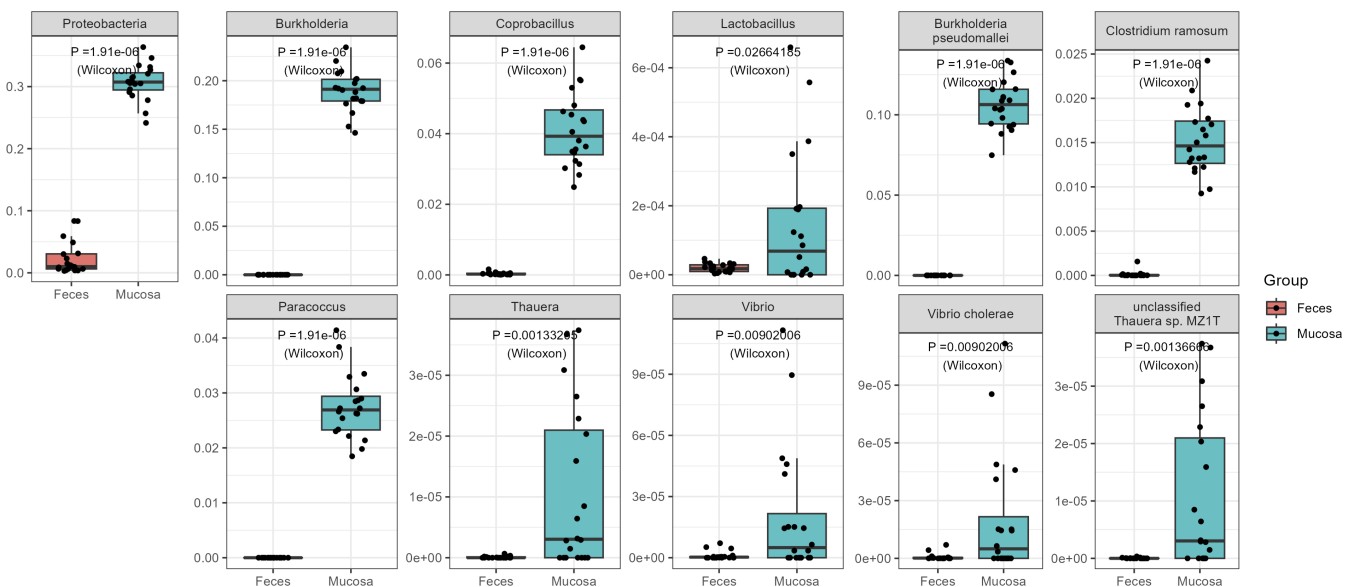

**FIG 3** Eleven differential biomarkers with significantly higher abundance in rectal mucosa than in feces at the phyla, genera, and species levels.

From functional and biological perspectives, the microbes abundant in the mucosa play diverse roles. The dominance of proteobacteria may be due to their adaptability to fluctuating oxygen levels and their association with inflammation, potentially serving as a diagnostic microbial marker for dysbiosis and disease risk (42). *Burkholderia*, especially *B. pseudomallei*, are opportunistic pathogens possibly exploiting compromised mucosal barriers (43, 44). *Coprobacillus*, identified as a potential biomarker in stool samples of CRC patients (45), suggests that its significance in mucosal health and has been spotlighted in CRC. *C. ramosum*, a general commensal, likely benefits from mucosal nutrients and is a common enteric anaerobe (46). Vibrio cholerae, a known enteropathogen (47), may preferentially adhere to and colonize the mucosal surface. *Lactobacillus*, with its probiotic properties, could enhance mucosal health by outcompeting potential pathogens and play a crucial role in gut health (48). Notably, *Paracoccus*, *Thauera*, and *unclassified Thauera sp. MZ1T* are novel findings in the mucosa, hinting at unique adaptations that merit further study and have not been reported in previous literature. These findings emphasize the significance of understanding the mucosal microbiome, offering insights into disease prevention, diagnostics, and potential therapeutic interventions.

The shotgun metagenome analysis offers insights into bacterial genomes which are beyond what 16S rRNA sequencing could afford. Based on the enrichment analysis using the reporter score algorithm (49), we identified KEGG Orthology (KO) pathways that were enriched in the mucosa compared with the feces, and vice versa (Fig. 4).

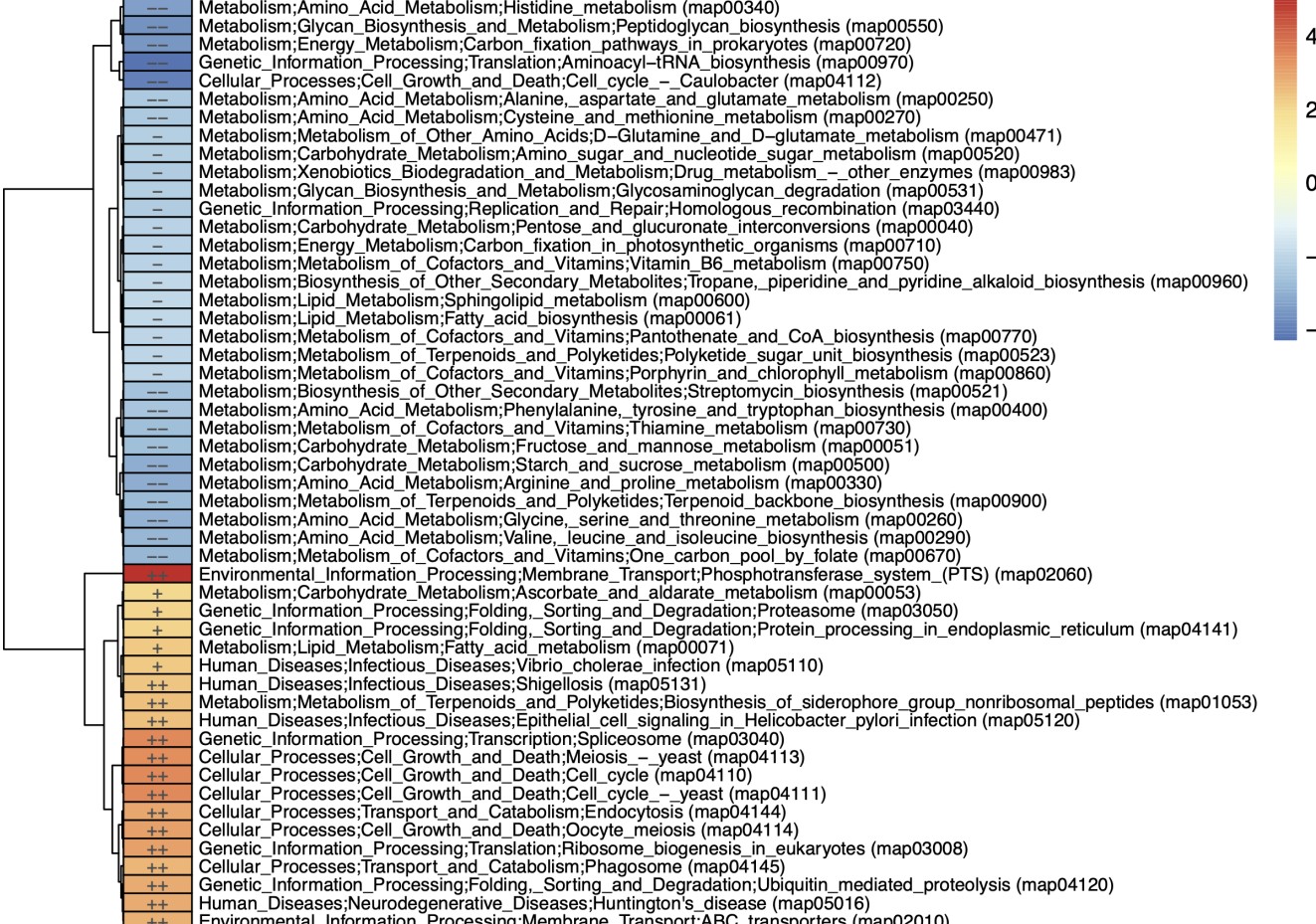

**FIG 4** Enriched KO pathways in feces and rectal mucosa, where red represents the mucosa group and blue denotes the feces group. --：reporter score <−2.3, -：reporter score <−1.96, ++：reporter score >2.3, +：reporter score >1.96.

Specifically, according to Fig. 4, we found that the top enriched pathway in the rectal mucosa is the phosphotransferase system (PTS), which is an important importer for mono- and di-saccharides (50). This suggests that the mucosa is a nutrient-rich environment, probably due to its proximity to mucus and gut epithelial cells, which are sources of host-derived carbohydrates such as fucose and sialic acids (51, 52). The fatty acid metabolism pathway is also enriched, as it is well known that some intestinal bacteria can produce a variety of short chain fatty acids through fermentation, and some can feed on these metabolites (53, 54).

In addition, we observed that two of the top enriched pathways in fecal microbiome are carbon fixation and cell cycle, suggesting that these microbes are metabolically active and proliferating. Interestingly, quite a few amino acid metabolism pathways are also enriched, including histidine, arginine, proline, glycine, and serine. This aligns with the fact that most proteolysis processes occur before food reaches the large intestine, and intestinal bacteria take advantage of free amino acids (55). The xenobiotic metabolism pathway is also enriched, which may be due to the fact the fecal microbiome originates from the gut lumen where they come into contact with host ingested drugs more frequently than mucosal microbiome that are shielded by mucus. Fatty acid synthesis pathway is enriched, which corresponds well with the earlier observation that the mucosal microbiome utilizes these metabolites, demonstrating a classic example of microbial cross-feeding in the intestinal ecosystem (56).

## The association between rectal mucosal microbiome and phenotypes

The gut microbiota is closely related to the systemic conditions and health-disease status of the host. Notably, due to the mucosal microbiota having an even more intimate interaction with the host, its composition or function is closely linked to the health of the host. To delve deeper into this relationship, we conducted further grouped statistical analyses based on the corresponding phenotypes and clinical information of the samples, such as age, gender, BMI, and polyp risk level.

While investigating the relationship between the rectal mucosal microbiome and age, we separated participants into two age groups, HIGH (54–68 years old) and LOW (44–53 years old), and looked for differential taxonomic features ($P < 0.05$, Fig. 5A). At phylum level, Firmicutes are enriched in the HIGH age group which is in accordance with previous studies (57). At genus level, both *Coprobacillus* and *Enterobacter* are enriched in the HIGH age group. Importantly, these findings are consistent with prior research based on feces, underscoring the overarching influence of age on the gut microbiota as a whole (58, 59). At the species level, we identified two differential microbes, a Firmicute *Clostridium ramosum* (*C. ramosum*) and a Proteobacterium *Enterobacter cloacae* (*E. cloacae*), that are enriched in the HIGH group. Notably, this is the first time these differential species have been reported given our study context. Intriguingly, the genera *Coprobacillus* and the species *C. ramosum* have recently been reported to be positively correlated with the severity of COVID-19 after SARS-CoV-2 infections (60). Combined with the novel findings of our study, it may be possible that the correlation exists due to their association with age, which is a prominent factor correlated with COVID-19 severity. It is important to emphasize that the aforementioned COVID-19 studies were also based on analyses of feces.

Similarly, when studying the relationship between the rectal mucosal microbiome and gender, we identified seven differential microbes that were more abundant in male subjects compared with female subjects ($P < 0.05$, Fig. 5B). These included two genera (*Blautia* and *Roseburia*) and five species (*Bacteroides intestinalis*, *Clostridium symbiosum*, *Eubacterium dolichum*, [*Ruminococcus*] *torques* and *unclassified Clostridium sp. HGF2*). The consistent observation of these microbes being more prevalent in male subjects prompts deeper reflection on the underlying factors and potential implications. Hormonal differences between genders, dietary habits, or other physiological and environmental factors specific to males might influence the abundance of these microbes (61, 62). Moreover, the male-predominant enrichment of these microbes could

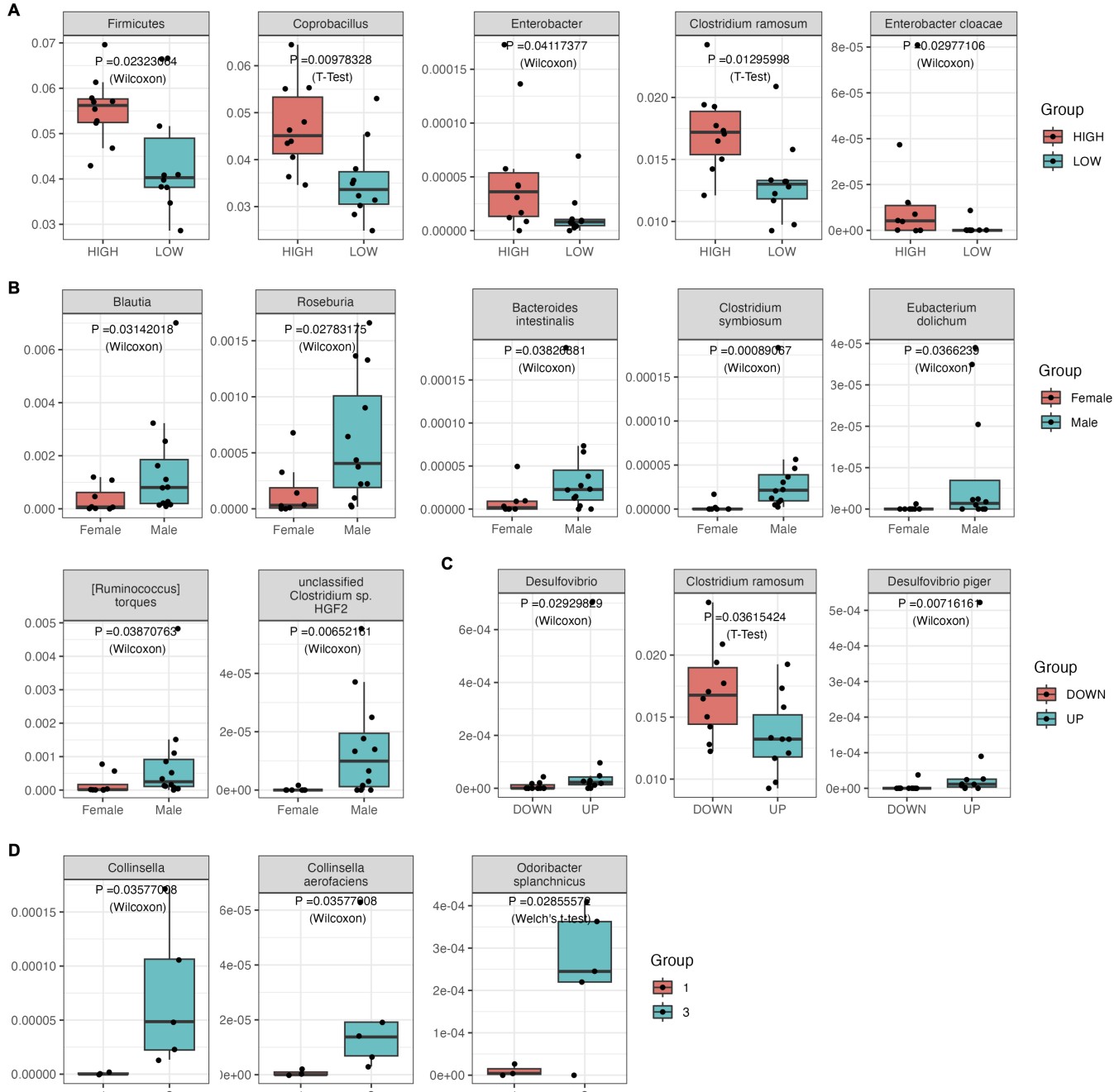

**FIG 5** Relative abundance of phenotype-associated differential biomarkers in the rectal mucosal microbiome at the phylum, genus, and species levels. (A) Age. (B) Gender. (C) BMI. (D) Polyp risk.

indicate a potential predisposition to certain health conditions or differential responses to environmental stimuli. This observation aligns with the broader understanding that gender can play a significant role in health outcomes and disease susceptibilities (63).

For instance, *Blautia*, a genus of anaerobic bacteria, produces butyric acid and acetic acid, compounds known to reduce obesity by regulating G-protein coupled receptors (GPRs) 41 and 43 (64). Cai et al. (15) reported a higher abundance of *Blautia* in the feces of colorectal neoplasm patients than in healthy individuals. Interestingly, while no significant gender differences were observed in tumor samples, our study found a higher prevalence of *Blautia* in male subjects (Fig. S2B). This is particularly noteworthy given

that previous research has established a correlation between *Blautia* and the severity of COVID-19 (65). Considering that gender, specifically being male, is also identified as a risk factor for COVID-19 (66), our findings hint at a potential interconnected role of gender and *Blautia* in disease susceptibility. This observation suggests that the gender-specific abundance of *Blautia*, as revealed in our study, finds indirect support from existing research, and emphasizes the importance of understanding gender-specific microbial profiles in the context of disease.

The other identified microbes, such as *Roseburia*, *B. intestinalis*, and *C. symbiosum*, also have significant roles in gut health and disease. For instance, *Roseburia* is a butyric acid-producing commensal bacterium essential for glycolipid metabolism (67), and *B. intestinalis* produces secondary bile acids among other metabolic functions (68). The consistent higher abundance of these microbes in male subjects raises questions about potential gender-specific health implications. Further research is imperative to elucidate the exact mechanisms and potential clinical implications of these gender-based microbial differences.

When investigating the relationship between the rectal mucosal microbiome and BMI, 20 samples were divided into two groups: the UP group (23.5–27) and the DOWN group (19.7–23). Three differential microbes were identified, including one genus (*Desulfovibrio*) and two species (*C. ramosum* and *Desulfovibrio piger*) ($P < 0.05$, Fig. 5C). However, for *Desulfovibrio*, the significance disappeared when the outlier sample was removed. The relative abundance of *C. ramosum* was significantly higher in the DOWN group than in the UP group, whereas the opposite was true for *D. piger*. *C. ramosum* is a Gram-positive, spore-forming, anaerobic bacterium that has been associated with obesity and metabolic syndrome symptoms in humans (69, 70). However, our results contradict this conclusion. A possible reason for this contradiction may be due to the studies being based on different types of samples, and further investigation is needed to clarify the impact of *C. ramosum* on obesity. Existing research suggests that the relative abundance of *D. piger* is significantly increased in overweight/obese patients from Sardinia compared with the normal-weight controls (71), which is consistent with our study's findings.

When exploring the relationship between the rectal mucosal microbiome and the risk of CRC (Fig. 5D), we categorized 20 paired samples into three groups based on the pathological risk level of the polyps: Group 1 (hyperplastic polyps, low risk, three samples), Group 2 (tubular adenomas, moderate risk, 12 samples), and Group 3 (tubulovillous adenomas, high risk, five samples). To avoid sample number bias, we focused on Group 1 and Group 3 for metagenomic differential analysis. Although there are only a small number of samples, our analysis revealed a significant enrichment of the genus *Collinsella* and two species, *Collinsella aerofaciens* and *Odoribacter splanchnicus*, in the rectal mucosa that is associated with the risk of CRC ($P < 0.05$, Fig. 5D), which were not observed in our feces data ($P > 0.1$, Fig. S2D).

*C. aerofaciens* is known as a beneficial commensal (72), producing short chain fatty acids, which were thought to play an important role in protecting against inflammation and CRC (73–75). However, our results contrast with this. One of the possible reasons for this discrepancy may be due to the fact that studies are based on different types of samples, and further research is needed to elucidate the relationship between *C. aerofaciens* and the risk of CRC. Consistent with previous studies, *O. splanchnicus* has been reported to be associated with promoting colorectal carcinogenesis, with its level being elevated in CRC patients (12). Furthermore, the relative abundance of these taxonomic groups was notably higher in Group 3 compared with Group 1 (Fig. 5D; Fig. S2D). An overall upward trend in the abundance of these taxa was observed across different risk levels in the rectal mucosal tissue. This trend was generally mirrored in feces, except for *O. splanchnicus* (Fig. S3).

Consistent with our findings, other studies have reported increased relative abundances of *Collinsella* in the gut microbiota of CRC patients compared with healthy controls (76, 77). *Collinsella* has also been proposed as a potential fecal marker for early

detection of CRC (76). These findings support our suggestion that mucosa samples can more directly identify differential microorganisms based on polyp risk levels.

## Pathway analysis based on phenotype features

We again performed KO pathway enrichment analysis on different phenotypes. The results showed significant differences in pathways among different comparison groups, including age, gender, and BMI (Fig. 6). This provides insights into the potential mechanisms underlying the development of intestinal diseases such as cancer and inflammatory bowel disease (IBD).

Three pathways (beta-lactam resistance, flagellar assembly and chemotaxis pathway) that play crucial roles in regulating various intestinal biological processes were found to be significantly enriched in the older age samples (HIGH group) with a reporter score of 2. The enrichment of beta-lactam resistance is probably due to accumulated exposure to antibiotics as a result of age. The two-cell motility-related pathway enrichments, namely flagellar assembly and chemotaxis, are an interesting observation and warrant further research into their mechanisms. In contrast, the enrichment of the nitrotoluene

**FIG 6** KO pathway enriched under different phenotypic groupings in rectal mucosa. (A) Age, where red represents the HIGH group. (B) Gender, where red represents the female group. (C) BMI, where red represents the UP group. --: reporter score <−2.3, -: reporter score <−1.96, ++: reporter score >2.3, +: reporter score >1.96.

degradation pathway suggests that, despite the apparent higher susceptibility to intestinal infections in younger individuals (LOW group), detoxification mechanisms can still protect them against intestinal diseases.

A gender bias in intestinal diseases, particularly IBD such as Crohn's disease and ulcerative colitis, is observed in the enriched pathways of cellular metabolism, signaling mechanisms, and neurodegenerative diseases in the female group. Previous studies have indicated that immune-mediated diseases typically show a female preponderance (78). In our analysis, the enrichment of pathways in females related to cell cycle and two-component systems suggests gender differences in immune cell activation and signaling (Fig. 6B). These differences could potentially influence the development and progression of IBD and cancer. On the other hand, the male group exhibits a preponderance (reporter score: −4) in protein synthesis, as the enrichment of ribosome, valine leucine and isoleucine biosynthesis pathways that can contribute to the functioning of cellular metabolism.

Additionally, a high BMI contributes to gut microbiota dysbiosis, dysregulation of cell growth and proliferation, and altered metabolism according to the significant enrichment of several pathways, including xenobiotic nitrotoluene and styrene degradation. In contrast, low BMI with an enrichment of lipid metabolism pathways and a replication and repair pathway, appears to have a lower risk of intestinal diseases.

## Conclusion

In this study, we utilized a unique approach by analyzing microbiome metagenomic sequencing results of paired normal rectal mucosa and feces, demonstrating our commitment to leveraging advanced sequencing techniques for deep insights into microbial composition of the mucosa. These findings not only shed light on the intricate relationship between the mucosal microbiome and human health but also revealed a new way for future research aimed at discovering novel biomarkers for human health and disease conditions.

Our study looked at the depth of the microbiome compositions of mucosa and feces from adults with colorectal polyps and identified differentiating features that related to age, gender, BMI, and polyp risk at different taxonomic levels (phylum, genus, species) as well as gene (pathway) level thanks to the metagenomic sequencing technology. It is worth noting that while our samples were collected during colonoscopy procedures for polyp removal, the normal rectal mucosa tissues were collected in our study to comprehensively describe the microbial characteristics of the rectal mucosa. Therefore, the results could be interpreted with an emphasis on the microbial landscape but not only the risk of colorectal polyps.

However, our study has certain limitations that need to be acknowledged, including the relatively small sample size and the absence of samples from lesions. Therefore, studies with larger cohorts and more comprehensive information are needed in the future.

## MATERIALS AND METHODS

### Patients and sample collection

We enrolled 20 patients who had colorectal polyps diagnosed by previous colonoscopies. These patients did not receive probiotics or antibiotics within the past 3 months. Feces were collected at home by these individuals before bowel cleansing and colonoscopy, and were sent by post to the research facility, where they were stored at −80°C. Normal rectal mucosal tissues were obtained from colonoscopies after bowel cleansing with 3-L-based PEG (polyethylene glycol)-based laxatives at Huazhong University of Science and Technology Union Shenzhen Hospital. One or two pinch biopsies (~ 3 × 3 mm) from the rectum were obtained using colonoscopic biopsy forceps, for metagenomic sequencing.

The study was approved by the local ethics committee of Huazhong University of Science and Technology Union Shenzhen Hospital, and informed consent was obtained from all patients before sample collection.

## DNA extraction

Microbiome DNA from feces and mucosa was extracted using the Omega E.Z.N.A. Stool DNA Kit (Omega Bio-tek, Inc., USA), following the manufacturer's instructions. The purity and integrity of the extracted DNA were analyzed using 1% agarose gel electrophoresis. The DNA concentration was precisely quantified using Qubit, and the samples were stored at −20°C for subsequent experiments.

## Library construction and sequencing

Qualified DNA samples were randomly fragmented into short segments of approximately 300 bp in length using the Covaris ultrasonic disruptor. The entire library preparation was completed through steps such as end repair, A-tailing, sequencing adapter ligation, purification, and PCR amplification. After the library construction was completed, an initial quantification was performed using Qubit 2.0, diluting the library to 2 ng/μL. Subsequently, the Agilent 2100 was used to detect the insert fragments of the library. Once the insert fragments met expectations, the effective concentration of the library was accurately quantified using the Q-PCR method (library effective concentration >3 nM) to ensure library quality. After the quality inspection of the library was qualified, different libraries were pooled based on effective concentration and the required sequencing data volume, and then sequenced on the Illumina NovaSeq PE150 platform, ensuring that the sequencing data volume for each sample was not less than 5G.

## Bioinformatics analysis of microbiota

The processing steps for the sequencing data obtained include (1) filtering out low-quality data, (2) aligning to the integrated gene catalog (IGC) gene set (79) using SOAP2 (80), and (3) based on the gene set, microbial classification, and the relationship with KEGG Orthology (KO), calculating the relative abundance of phylum, genus, species, and KO. The species diversity between samples (Beta diversity) was represented by the Bray-Curtis distance. Principal Coordinate Analysis (PCoA) is used to display the Bray-Curtis distances between samples.

## Statistical analysis

All statistical analyses were performed using Python (version 3.6.13) and R (version 3.6.1). Most of the data processing was done by Python package pandas. Graphical visualizations were generated by the R package ggplot2. Considering that our samples consist of paired data, we ultimately chose to use paired $t$-tests or Wilcoxon signed-rank tests to analyze the differential abundance of microbes between different groups. $P < 0.05$ was considered statistically significant. Based on the relative abundance of KO in the samples, the z-score of KO pathways was calculated using the reporter score algorithm (49). When the absolute value of the z-score is greater than 1.96 (97.5% confidence level of the normal distribution), the result is considered to have a significant difference between different groups.

## ACKNOWLEDGMENTS

This research was funded by grants from the Shenzhen Nanshan District Scientific Research Program of the People's Republic of China, grant number 2020001. Additionally, the project was funded by the Science and Technology Development Fund, Macau SAR (File no. 006/2023/SKL).

Conceptualization, X.Y. and H.C.; Formal analysis, T.Y.; Funding acquisition, X.Y.; Investigation, X.M.; Methodology, T.Y.; Project administration, H.C. and W.K.; Resources,

X.Y., H.C. and Y.H.; Software, T.Y.; Supervision, X.Y., T.Y. and H.C.; Validation, T.Y. and J.L.; Visualization, T.Y. and J.L.; Writing – Original draft, X.Y., T.Y. and J.L.; Writing – Review & Editing, X.Y., T.Y., J.L., H.L., J.W. and H.C.

## AUTHOR AFFILIATIONS

[1]Department of Gastroenterology, Huazhong University of Science and Technology Union Shenzhen Hospital, The 6th Affiliated Hospital of Shenzhen University Health Science Center, Shenzhen, China
[2]iCarbonX(zhuhai) Company Limited, Zhuhai, China
[3]Shenzhen Digital Life Institute, Shenzhen, China
[4]State Key Laboratory of Quality Research in Chinese Medicines, Macau Institute for Applied Research in Medicine and Health, Macau University of Science and Technology, Macau, China
[5]Department of Digital Health, South China Hospital of Shenzhen University, Shenzhen, China

## AUTHOR ORCIDs

Xiao-Fei Yin  http://orcid.org/0000-0001-7656-9007
Hongzhi Cao  http://orcid.org/0000-0003-3658-9884
Wen-Quan Kang  http://orcid.org/0009-0005-1565-4909

## FUNDING

| Funder | Grant(s) | Author(s) |
|---|---|---|
| Shenzhen Nanshan District Scientific Research program of the People's Republic of China | 2020001 | Xiao-Fei Yin |
| The Science and Tchnology Development Fund, Macau SAR | 006/2023/SKL | Jun Wang |

## DATA AVAILABILITY

All raw shotgun data have been deposited in the European Nucleotide Archive (ENA) with the BioProject ID PRJEB67450, under the accession number ERA27310965.

## ADDITIONAL FILES

The following material is available online.

### Supplemental Material

**Fig. S1 (Spectrum03549-23-s0001.tif).** Detected microbial genes.
**Fig. S2 (Spectrum03549-23-s0002.tif).** Relative abundance of phenotype-associated differential biomarkers.
**Fig. S3 (Spectrum03549-23-s0003.tif).** Relative abundance distribution of the three taxonomic groups.
**Supplemental material (Spectrum03549-23-s0004.docx).** Legends for Fig. S1 to S3.
**Supplemental tables (Spectrum03549-23-s0005.xlsx).** Tables S1 to S4.

### Open Peer Review

**PEER REVIEW HISTORY (review-history.pdf).** An accounting of the reviewer comments and feedback.

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
