## [Reviewer comments · Microbiology Spectrum]

Microbiology Spectrum

The microbiome compositional and functional differences between rectal mucosa and faces

xiaofei Yin, taoyu Ye, hanlin chen, junyan Liu, xuefeng Mu, hao Li, Jun Wang, yuanjia HU, hongzhi Cao, and wenquan Kang

Corresponding Author(s): wenquan Kang, Department of Gastroenterology, Huazhong University of Science and Technology Union Shenzhen Hospital, The 6th Affiliated Hospital of Shenzhen University Health Science Center, Shenzhen, China

Review Timeline:

Submission Date:	October 9, 2023
Editorial Decision:	February 11, 2024
Revision Received:	March 26, 2024
Accepted:	May 6, 2024

Editor: Jinxin Liu

Reviewer(s): Disclosure of reviewer identity is with reference to reviewer comments included in decision letter(s). The following individuals involved in review of your submission have agreed to reveal their identity: Hao-Yu Liu (Reviewer #2)

Transaction Report:

DOI: <https://doi.org/10.1128/spectrum.03549-23>

Re: Spectrum03549-23 (The microbiome compositional and functional differences between rectum mucosa and faeces)

Dear Mr. wenquan Kang:

Thank you for the privilege of reviewing your work. Below you will find my comments, instructions from the Spectrum editorial office, and the reviewer comments.

Revision Guidelines

Sincerely,
Jinxin Liu
Editor
Microbiology Spectrum

Reviewer #1 (Comments for the Author):

In this study, the authors collected 20 patients with colonic polyps to compare the microbial communities between rectal mucosa and feces. They found that the biodiversity in feces is higher and many microbes found in the mucosa can also be detected in feces. From their results, Burkholderia genus is the most relevant discriminating factor between feces and mucosa. This research provided comprehensive results that fecal sample may somehow reflect the microbial communities of rectal mucosa.

Reviewer #2 (Comments for the Author):

The manuscript 'The microbiome compositional and functional differences between rectum mucosa and faeces' has collected colonic polyps and fecal samples from 20 patients and investigated the microbial compositional and functional differences between sample types using shotgun metagenomics, therefore providing insights into the more precise role of gut microbiota in the colon cancer development. It is well-designed and the analysis is thorough and indeed is relevant to the field. However, there are several issues needed to be addressed.

1. First and foremost, in the results section, there was large part of discussion mixed, without proper citation. My suggestion would be just merging results and discussion AS RESULTS AND DISCUSSION, and have the limitation of the study and conclusion as a separate section, ie., CONCLUSION/CONCLUDING REMARKS etc. below are places need proper citations, it is probably more than what I've listed, please go through the body text:
Line 134-137, citation; Line 168, citation; Line 192-196, need citation; Line 233 citation; Line 235-240 citation; Line 245-251, citation; Line 281-285 citation; Line 381, please do not call out results in discussion, and citation.
2. Regarding the Abstract, please check your first paragraph of discussion, include more specific information in the abstract, for instant, you found that 'the Burkholderia genus is the most relevant discriminating factor between faeces and mucosa.'
3. Please check the whole manuscript including figures and chose to use 'feces' or 'faeces', be consistent.
4. For figure 1B, could you use color for sampling sites as it is and perhaps use different shapes for different sex? Since gender-specific effects were discussed later in the paper.
5. I did not find accession number for your metagenomic data, please provide it.
6. The authors mentioned that most studies examined fecal microbiota, which make their study very unique. I think it is very interesting. Meanwhile, are there studies comparing cancer-associated microbiota, feces versus mucosa in mice? Please try to include and discuss what's known and comparable to your study, if there are studies available.

May 6, 2024

Dear **Dr. Jinxin Liu**

Thank you for reviewing our manuscript entitled “The microbiome compositional and functional differences between rectal mucosa and feces” (Paper #Spectrum03549-23). After fully considering the opinions of reviewers and making appropriate and comprehensive revisions to the manuscript, we believe it will be of considerable interest to the broad readership of Microbiology Spectrum. We are hereby resubmitting the revised manuscript, which has been carefully revised according to the reviewers’ comments and suggestions. Point-by-point responses to the comments are appended, as well as the revised manuscript.

The authors appreciate your consideration of our manuscript again, and we look forward to hearing from you soon.

Sincerely yours,

Xiaofei Yin

Email: yinx0901@126.com

Response to reviewers

Response to Reviewer #1

Comments:

Reviewer #1: In this study, the authors collected 20 patients with colonic polyps to compare the microbial communities between rectal mucosa and feces. They found that the biodiversity in feces is higher and many microbes found in the mucosa can also be detected in feces. From their results, Burkholderia genus is the most relevant discriminating factor between feces and mucosa. This research provided comprehensive results that fecal sample may somehow reflect the microbial communities of rectal mucosa.

Response: thanks very much for the reviewer's comment on our manuscript.

Response to Reviewer #2

Comments:

Reviewer #2: The manuscript 'The microbiome compositional and functional differences between rectum mucosa and feces' has collected colonic polyps and fecal samples from 20 patients and investigated the microbial compositional and functional differences between sample types using shotgun metagenomics, therefore providing insights into the more precise role of gut microbiota in the colon cancer development. It is well-designed and the analysis is thorough and indeed is relevant to the field. However, there are several issues needed to be addressed.

(1) First and foremost, in the results section, there was large part of discussion mixed, without proper citation. My suggestion would be just merging results and discussion AS RESULTS AND DISCUSSION, and have the limitation of the study and conclusion as a separate section, ie., CONCLUSION/CONCLUDING REMARKS etc. below are places need proper citations, it is probably more than what I've listed, please go through the body text:

Line 134-137, citation; Line 168, citation; Line 192-196, need citation; Line 233 citation; Line 235-240 citation; Line 245-251, citation; Line 281-285 citation; Line 381, please do not call out results in discussion, and citation.

Response: Many thanks for your detailed feedback. **Following your suggestion, we have carefully addressed the mixed discussion within the Results section by ensuring proper citations both at the listed lines and throughout the manuscript wherever similar issues were identified. Furthermore, following your valuable**

advice, we have merged the Results and Discussion sections into one comprehensive “RESULTS and DISCUSSION” section. We have also created a separate section for “CONCLUSION”, clearly delineating the study’s limitations and summarizing our findings. This restructuring enhances the clarity and coherence of our manuscript, and we believe it significantly improves the presentation and interpretation of our research. Please find the corresponding change at Line: 136–139; Line: 165; Line: 186–190; Line: 220–226, Line: 231–238; Line: 268–275, etc.

(2) Regarding the Abstract, please check your first paragraph of discussion, include more specific information in the abstract, for instant, you found that 'the Burkholderia genus is the most relevant discriminating factor between faeces and mucosa.'

Response: Many thanks for your comments. Following your suggestion, we have modified our Abstract and added more specific information. The main revised Abstract is:

Our findings revealed a pronounced distinction of the microbial communities between these two sample sets. Compared to feces, the mucosal microbiome contains fewer genera, with *Burkholderia* being the most discriminating genus between feces and mucosa, highlighting its significant influence on the mucosa. Furthermore, based on the microbial classification and KEGG Orthology (KO) annotation results, we explored the association between rectal mucosal microbiota and factors such as age, gender, BMI, and polyp risk level. Notably, we identified novel biomarkers for these phenotypes, such as *Clostridium ramosum* and *Enterobacter cloacae* in age. The mucosal microbiota showed an enrichment of KO pathways related to sugar transport and short chain fatty acid metabolism. Our comprehensive approach not only bridges the knowledge gap regarding the microbial community in the rectal mucosa but also underscores the complexity and specificity of microbial interactions within the human gut, particularly in the Chinese population.

(3) Please check the whole manuscript including figures and chose to use 'feces' or 'faeces', be consistent.

Response: Thank you for your comments. Following your comments, we have carefully reviewed the entire manuscript, including figures, and have ensured consistent use of the term 'feces' throughout.

(4) For figure 1B, could you use color for sampling sites as it is and perhaps use different shapes for different sex? Since gender-specific effects were discussed later in the paper.

Response: Many thanks for your suggestion. According to your comments, we have revised Figure 1B accordingly. We now use colors to represent the sampling sites

as originally intended and have introduced different shapes to denote gender: circles for females and triangles for males. Figure 1 is shown below:

Figure 1. Sampling locations and general overview of the fecal and mucosa-derived metagenomic datasets. (A) For each of the 20 enrolled subjects, biopsies of normal gut mucosa were collected from the rectum, and feces were also sampled. (B) PCoA plots of the downsized mucosa-feces dataset at the phylum level, color-coded by sampling-location and shape-coded by gender. (C) Information on the genera of feces and mucosa.

(5) I did not find accession number for your metagenomic data, please provide it.

Response: Thank you very much for your comments. **According to your comments, we have added the accession number for our metagenomic data in “4.6 Data availability” section.** The corresponding information is:

All raw shotgun data have been deposited in the European Nucleotide Archive (ENA) with the BioProject ID PRJEB67450, under the accession number ERA27310965.

(6) The authors mentioned that most studies examined fecal microbiota, which make their study very unique. I think it is very interesting. Meanwhile, are there studies comparing cancer-associated microbiota, feces versus mucosa in mice? Please try to include and discuss what's known and comparable to your study, if there are studies available.

Response: Many thanks for your insightful comment. In response to your suggestion, we conducted a systematic search on PubMed using the following keywords: (microbiome OR microbiota) AND (feces OR fecal) AND (mucosa OR mucosal) AND (mice OR mouse) AND (colorectal cancer OR CRC OR colorectal tumor) AND (difference). The aim was to identify studies that directly compare cancer-associated microbiota in feces versus mucosa in mice and its relevance to our study's focus. Our review of the titles and abstracts of the retrieved studies did not find any research that compares these aspects with direct relevance to our study. However, we did identify one indirectly related study. Saffarian et al. (1) investigated the cypt- and mucosa-associated core microbiotas in humans and their alteration in colon cancer patients by utilizing laser microdissected tissues for 16S rRNA gene sequencing. This

study provides valuable insights into microbiome's role in colorectal cancer, albeit with a focus on human subjects rather than mice.

Furthermore, we performed further exploratory research using Google Scholar and Bing to retrieve relevant studies conducted on mice. For example, Zhang et al. (2) explored the effects of rifaximin on the composition of ileal, colonic mucosal, and fecal microbiota in mice with post infectious irritable bowel syndrome (PI-IBS) using 16S rRNA sequencing. Wong et al. (3) determined that fecal microbiota from patients with colorectal cancer (CRC) can promote tumorigenesis in germ-free mice and mice given a carcinogen by fed stool samples from CRC patients and healthy individuals, collection of intestinal tissues for histological evaluation, and performing 16S rRNA gene sequencing analysis of feces.

These studies provide valuable insights into the relationship between the microbiota of different gut compartments and disease states, underscoring the significance of our findings within the broader research field. However, it is important to note that these studies all used 16S sequencing methods, while metagenomic sequencing was used in our study. In the future, we will continue to monitor the latest research developments in this field.

Reference

1. Saffarian A, Mulet C, Regnault B, Amiot A, Tran-Van-Nhieu J, Ravel J, Sobhani I, Sansonetti PJ, Pédrón T. 2019. Crypt- and Mucosa-Associated Core Microbiotas in Humans and Their Alteration in Colon Cancer Patients. *mBio* 10:e01315-19.
2. Zhang S, Hong G, Li G, Qian W, Jin Y, Hou X. 2023. Modulation of the microbiota across different intestinal segments by Rifaximin in PI-IBS mice. *BMC Microbiol* 23:22.
3. Wong SH, Zhao L, Zhang X, Nakatsu G, Han J, Xu W, Xiao X, Kwong TNY, Tsoi H, Wu WKK, Zeng B, Chan FKL, Sung JJY, Wei H, Yu J. 2017. Gavage of Fecal Samples From Patients With Colorectal Cancer Promotes Intestinal Carcinogenesis in Germ-Free and Conventional Mice. *Gastroenterology* 153:1621-1633.e6.

Re: Spectrum03549-23R1 (The microbiome compositional and functional differences between rectal mucosa and faces)

Dear Mr. wenquan Kang:

Your manuscript has been accepted, and I am forwarding it to the ASM production staff for publication. Your paper will first be checked to make sure all elements meet the technical requirements. ASM staff will contact you if anything needs to be revised before copyediting and production can begin. Otherwise, you will be notified when your proofs are ready to be viewed.

Sincerely,
Jinxin Liu
Editor
Microbiology Spectrum

Reviewer #1 (Comments for the Author):

I don't have further questions.